# Comparison of Self-Reported Speed of Eating with an Objective Measure of Eating Rate

**DOI:** 10.3390/nu12030599

**Published:** 2020-02-26

**Authors:** Eilis Woodward, Jillian Haszard, Anna Worsfold, Bernard Venn

**Affiliations:** Department of Human Nutrition, University of Otago, P.O. Box 56, Dunedin 9016, New Zealand; EilisRose.Woodward@middlemore.co.nz (E.W.); jill.haszard@otago.ac.nz (J.H.); annaj1101@icloud.com (A.W.)

**Keywords:** eating rate, speed of eating, self-reported eating rate, objective eating rate

## Abstract

Slow eating may be beneficial in reducing energy intake although there is limited research quantifying eating rate. Perceived speed of eating was self-reported by 78 adults using a standard question “On a scale of 1–5 (very slow–very fast), how fast do you believe you eat?” Timing the completion of meals on three occasions was used to assess objective eating rate. The mean (SD) speeds of eating by self-reported categories were 49 (13.7), 42 (12.2), and 35 (10.5) g/min for fast, medium, and slow eaters, respectively. Within each self-reported category, the range of timed speed of eating resulted in considerable overlap between self-identified ‘fast’, ‘medium’ and ‘slow’ eaters. There was 47.4% agreement (fair) between self-reported speed of eating and the objective measure of eating rate (κ = 0.219). Self-reported speed of eating was sufficient at a group level to detect a significant difference (10.9 g/min (95% CI: 2.7, 19.2 g/min, *p* = 0.009)) between fast and slow; and fast and medium eaters (6.0 g/min (0.5, 11.6 g/min *p* = 0.033)). The mean difference (95% CI) between slow and medium eaters was 4.9 (−3.4, 12.2) g/min (*p* = 0.250). At an individual level, self-report had poor sensitivity. Compared to objectively measured speed of eating, self-reported speed of eating was found to be an unreliable means of assessing an individual’s eating rate. There are no standard protocols for assessing speed of eating or eating rate. Establishing such protocols would enable the development of population reference ranges across various demographic groups that may be applicable for public health messages and in clinical management.

## 1. Introduction

The rate at which a meal is eaten has been found to affect gut hormones associated with satiety signals, with slower eating leading to a more anorexigenic response compared with faster eating [1]. Eating too quickly for the response of satiety signals may explain why faster eating has been associated with increased energy intake in both observational and experimental studies [1,2,3,4,5]. With fast eating being positively associated with energy intake, an association between eating rate and body mass index (BMI), obesity and metabolic disease has also been found [6,7,8,9,10,11,12,13,14,15,16]. Promoting a slower eating rate may therefore be advisable to reduce health risks. However, there is no fixed frame of reference or any clinically relevant guidance for defining an appropriate objective eating rate, with calls for a more systematic approach in the assessment of eating behaviours on health outcomes [17].

In order to advise people of healthful eating rates, it is important to establish a means of identifying fast eaters. One common indicator used in research is to ask people whether they are fast or slow eaters, using five self-reported categories: very slow, relatively slow, medium, relatively fast, and very fast [9,11,18,19,20,21,22,23,24]. Self-reported speed of eating, however, is a highly subjective perception, which also does not have a frame of reference by which to compare against an objective measure.

In 2003, Sasaki and colleagues assessed the validity of self-reported speed of eating at an individual level. As a group, in 222 subjects there was 46% exact agreement between friend- and self-reports [24]. However, at an individual level using a subjective measure (friend-report) was insufficient to validate self-report speed of eating. Although less than 50% correspondence is not ideal, many people have used these results to justify the use of self-reported speed of eating [21,25,26]. 

Various methods have been used to measure objective eating rate in the home and laboratory. Videotaping has been used as a means of covertly monitoring eating rate so as not to interfere with normal eating behaviour [27]. In a laboratory setting, electronic weighing scales connected to computers have been concealed in bench tops enabling eating rates to be covertly measured throughout the duration of a meal [28]. The overall time to ingest a fixed amount of food has also been used to assess eating rate, both overtly by asking participants to self-time [29] and covertly [30]. Some work has been conducted in assessing intra-individual variability, with eating rate found to be ‘relatively’ consistent across four meals [31].

Thus, although eating rates are measurable, there are no population reference ranges by which to establish a norm and comparisons have not been made at an individual level between a person’s perception of self-reported speed of eating and their objectively measured eating rate. Therefore, the primary aim of this observational study is to investigate the agreement between self-reported speed of eating and eating rate measured objectively in the laboratory.

## 2. Materials and Methods 

### 2.1. Study Design

The study involved participants self-reporting their speed of eating and an experimental phase in which participants were covertly timed while eating three lunch meals designed by dietetic students that were provided one week apart. Participants were blinded to the purpose of the experiment, understanding that the purpose was to rate meal satisfaction. As such, participants were unaware that they were being timed and filled out a meal satisfaction questionnaire at the completion of each meal. Participants were randomised to receive either rice- or pasta-based meals unless a person indicated that he/she was gluten intolerant, in which case rice was specified. Participants were male and female students aged 18–60 years. Recruitment was via verbal and email invitation to a third-year class of students of human nutrition. Participation was voluntary with inclusion requiring written consent. People with self-reported food allergies to any of the study foods were excluded. Demographic information was collected via a paper survey and participant’s weights and heights were measured using a stadiometer and electronic scales. Body mass index was calculated as weight (kg) divided by height (m) squared. The study gained ethical approval from The University of Otago Human Ethics Committee.

### 2.2. Self-Reported Speed of Eating

Information was obtained via an eating habits questionnaire comprising questions regarding usual eating frequency in a day, usual evening meal environment, usual hunger and satiety, food allergies, and speed of eating. Questions other than speed of eating were asked to draw participant attention away from the primary research aim and these ancillary questions were not analysed. Self-reported speed of eating was based on a question validated by Sasaki et al. [24] in which participants were asked; ‘On a scale of 1–5, how fast do you believe you eat?’ with options; ‘very slow’, ‘relatively slow’, ‘medium’, ‘relatively fast’, and ‘very fast’ [24]. The questionnaire was paper based and completed by all participants one week before the experimental phase. 

### 2.3. Study Meals

The size of the meals was pre-tested by three female dietetic students. The criterion was that the meals should be of a size that could be comfortably eaten by female participants. The meals were prepared on site and contained starchy carbohydrate (rice or pasta) and non-starchy vegetables together with beef mince and bolognese sauce. Jasmine rice (Pams®; Auckland, New Zealand) was cooked in a rice cooker with a 1:2 ratio by weight of rice to water. Penne pasta (San Remo, San Remo Macaroni Co Pty Ltd; Adelaide, Australia) was added to boiling water and cooked until al dente, after which it was rinsed with cold water and tossed with 2 tablespoons of canola oil (Sunfield Oils, Tasti Products Ltd, Auckland; New Zealand) per 2.5 kg dried pasta. Asian stir-fry vegetables (Watties®, Heinz Watties Ltd, Auckland, New Zealand) were microwaved in 750 g portions for 10 minutes. One-kilogram batches of beef mince were browned with 2 teaspoons of canola oil and 1 kg of bolognaise sauce added (Dolmio®, Mars Food; Ballarat, Australia). The meals weighed 550 g and the composition of the three meals differed in the proportion of starchy carbohydrate and non-starchy vegetables as given in Table 1. 

The energy content of the meals was estimated using manufacturers nutrient information. The order in which participants received the meals was randomised. After finishing each meal participants provided feedback via a paper survey consisting of three 100 mm visual analogue scales (VAS) to assess post-prandial comfort, meal size, and taste. Comfort and meal size scales were anchored with the descriptors ‘too small’ to ‘too big’; with the taste scale anchored with ‘disliked’ and ‘liked a lot’. ‘Just right’ corresponded to the mid-point of the scales.

### 2.4. Objective Eating Rate

Participants arrived at the laboratory having fasted for a minimum of 2.5 hours. The meals were served in two sittings, with participants allocated a start time of either 11:50 or 13:00. For each participant, the start time was the same for all three meals. Additionally, participants were block randomised by sex to the order in which they received the meals (either rice or pasta based) in a balanced manner such that there were approximately equal numbers of six meal order permutations. The laboratory was spacious, allowing participants to be seated at least 1.5 metres apart from each other. Participants were permitted to talk to each other but reading material, computers and cell phones were not to be used whilst the meals were being consumed. The meals were served warm and participants were given verbal instructions to eat the entire meal (550 g) and to drink a 250 mL glass of water in a manner consistent with how they would normally consume foods and beverages. Research staff were allocated five to seven participants per session to covertly record the start and finish times via a digital stopwatch, one stopwatch for each participant. Research staff were sat at desks offering clear visibility of their allocated participants with stopwatches concealed behind an upstand. The start and stop times were defined as when food first entered a participant’s mouth to when a participant swallowed their last mouthful. The participants were unaware that they were being timed.

### 2.5. Statistical Analysis

Eating rate was defined as the weight of meal (550 g) divided by the time taken to eat the meal, expressed as g/min. An average eating rate was calculated for each person by taking the mean eating rate of the three meals. Within-person standard deviations and coefficient of variations were also calculated to assess within-person variability of objective eating rate. Mixed-effects regression analysis was used to determine mean differences in objective eating rate among the speed of eating categories. The model included the three objective eating rates for each participant and was adjusted for type of meal (rice or pasta), randomised order and had participant as a random effect. 

Objective eating rates were ranked and then divided into groups of slow, medium, and fast based on the number of participants for each self-reported speed of eating category. The speed of eating categories was compared against the objective groups for level of agreement using Cohen’s Kappa Coefficient and percent agreement. Consistency of objective eating rate was assessed with an intraclass correlation coefficient, using a two-way mixed-effects model. Data were analysed using Stata 13.1 (StataCorp, College Station, TX, USA) and Microsoft Excel 2011.

## 3. Results

Participants were young and healthy, predominantly of New Zealand European descent, and were mostly female. All 78 participants consumed all three meals and completed the pre-intervention eating habits questionnaire. On average, the meal with the smallest portion of rice or pasta, corresponding to the largest amount of non-starchy vegetables, was rated 61/100 for comfort and size and 48/100 for taste. The majority (68%) of the sample self-identified as New Zealand European. There were four Māori, eight Chinese and 12 people of miscellaneous ethnicity. Seventy-five percent of the sample were female. The mean (SD) age of the sample was 21.4 (2.1) y. Baseline characteristics of the study population categorised by speed of eating are presented in Table 2. The fast category included participants who identified as fast eaters (*n* = 35) and very fast eaters (*n* = 2). The slow category included participants who identified as slow eaters (*n* = 9) and very slow eaters (*n* = 1).

### 3.1. Objective Eating Rate

The median objective eating rates ranged from 33 g/min in the slow eaters group, to 50 g/min in the fast eaters group. There was moderate variation in objective speed of eating across the three meals, with the median coefficient of variation sitting at approximately 13% (Table 2). The intraclass correlation coefficient for the three different meals was 0.90. On average, the mean objective eating rate increased by 5.6 g/min with each category (*p* for trend *=* 0.003). The mean difference (95% CI) between the average speed of self-reported fast and slow eaters was 10.9 (2.7, 19.2) g/min (*p =* 0.009) and between fast and medium eaters was 6.0 (0.5, 11.6) g/min (*p =* 0.033). The mean difference (95% CI) between slow and medium eaters was 4.9 (-3.4, 12.2) g/min (*p =* 0.250). 

### 3.2. Agreement between Self-Reported Speed of Eating and Objective Eating Rates

The eating rates among the self-reported speed of eating categories are shown in Figure 1. The data represent the mean eating rate of the three meals for each person that have then been classified according to self-reported speed of eating. 

There was considerable overlap between the slow/medium and the fast/medium categories. There was even some overlap in eating rate between the people who self-classified themselves as having a slow or fast speed of eating.

Objective eating rates were ranked and split into three groups to match the group-size of self-reported eating groups, and labelled slow, medium, and fast. The agreement between self-reported speed of eating and objective eating rate is reported in Table 3. By groups, there was 47.4% agreement between speed of eating and eating rate. In accordance with the classification assigned to Cohen’s kappa coefficient there was ‘fair’ agreement between speed of eating and eating rate (κ = 0.219) [32]. 

Of those that classified themselves as having a slow speed of eating (*n* = 10), 40% (4 out of 10) were also categorized as having a slow objective eating rate. The corresponding agreement for people classifying as having a medium and fast speed of eating were 39% (*n* = 31) and 57% (*n* = 37), respectively. 

## 4. Discussion

Overall approximately half of the sample considered themselves to be fast or very fast eaters, 40% medium and 13% slow or very slow. At a group level, the speed of eating categories corresponded to differences in the objectively measured eating rate. However, at an individual level, there was considerable overlap of the measured eating rate among the speed of eating categories, resulting in only fair agreement between the subjectively and objectively determined data. While there was some variation between meals for objective eating rate, this was relatively low, with coefficient of variations (CVs) of approximately 13%. Furthermore, consistency between measures of objective eating rates were high, with an intra-class correlation (ICC) of 0.90, which is similar to those found in another study [31].

Self-reported speed of eating has been positively associated with disease risk in middle-aged people [15,33]. Such associations indicate the value of self-report as a predictor of risk, but little work has been carried out to assess how a person’s perception of speed of eating correlates with an objectively measured eating rate. Indeed, there is no standard by which a person is able to quantify their speed of eating other than by observing the eating habits of others. This may leave speed of eating susceptible to poor sensitivity and specificity. In a large group (*n* = 1695) of 18-year-old female Japanese students, approximately one-third of the women self-classified their speed of eating as slow (27%), medium (36%) and fast (37%) [24]. In our group of predominantly female students of comparable age, just 13% of our sample self-identified as having a slow speed of eating. The lower proportion of slow speed of eating is more comparable to that found in an adult Chinese population (*n* = 7972), in which males and females self-classified as slow, medium and fast eaters in the proportions of 8%, 49% and 43% (males) and 15%, 56% and 29% (females), respectively [33]. Differences in the proportion of slow, medium and fast speed of eating among studies may simply be differences in the characteristics of the samples or indicative of the arbitrary nature of self-perception of speed of eating. In the absence of a reference speed, self-perception is imprecise and likely to lead to some misclassification.

Another way of assessing speed of eating is to objectively measure eating rate by monitoring the time taken for participants to eat food. This has been done both overtly with participants aware that they are being timed [29], or covertly [30]. A group of obese youth eating hospital meals had a mean initial eating speed of 29.8 g/min that reduced to 26.4 g/min after training to slow down, a strategy that the authors recommended to induce ‘satiety responsiveness’ [34]. These eating rates already fall largely within our slow group (median 33 g/min), with our fast group having a considerably larger eating rate of 50 g/min. Nevertheless, eating rates vary widely with rates dependent upon the type of food being consumed. The rate at which young men ate a chocolate pudding decreased from a mean of 104 g/min to 71 g/min by increasing its viscosity [35]. Similarly, changing the texture of porridge from thick/more chewy to thin/less chewy resulted in eating rates of 74 g/min and 140 g/min, respectively [36]. There is no reference meal by which to establish a standard eating rate for a person, so investigators have compared self-reported speed of eating to objectively measured eating rates.

A difference in the eating rate of a pasta meal was found between fast (83 kcal/min) and slow (53 cal/min) speed of eating groups in 60 university students [22]. Similarly, people identifying as having a fast or very fast speed of eating ate a rice ball in less time than people who considered themselves to have a medium or slow speed of eating [18]. The data are consistent with these earlier reports as the mean objective eating rate differed across the self-report speed of eating categories. These findings suggest that self-reported speed of eating is predictive of actual eating rate at a group level, most consistently when comparing between fast and slow speed of eating. Additionally, we assessed the agreement between self-reported speed of eating and objective eating rate and found this to be only fair, lacking sensitivity and specificity. Thus, it appears that asking people about their speed of eating is useful in the identification of group-level differences in eating rate, but less so for individuals. The lack of a reliable and consistent measure of ascertaining subjective speed of eating or an objective test of eating rate is problematical because retraining eating behaviour at an individual level has been found to have health benefits [37]. The need for retraining may be missed if a person with a fast eating rate self-reports their speed of eating as slow or medium. A more definitive measure as to a person’s eating habits would be to develop a standardized meal and to record the time taken to eat this meal.

A limitation of our work may be that we assessed speed of eating by asking the participants directly whereas others have also asked friends to comment on the participant’s speed of eating [24,26]. However, the exact and adjunct agreement between speed of eating and eating rate was similar between our data and studies in which friend report had been used. Another limitation is the use of a laboratory, a setting that differs from eating at home. We are unaware of studies in which comparisons have been made between eating rates measured at home and in a laboratory, although caloric intake was found to be unaffected by a home or laboratory setting [38]. To what extent the setting affects eating rate is unknown. Laboratory conditions can be controlled for comparative purposes whilst many of our participants reported eating dinner at home with distractions such as television and other devices. Distracted mealtimes have been associated with impaired recollection of the meal consumed [39,40,41,42,43]. Other limitations are that the meals may not have represented foods and amounts normally consumed by the participants. The meals were designed such that people felt comfortable eating the entire meal. From feedback no-one indicated that the meals were difficult to finish and two of the males commented that they could eat more. The meal choice of rice or pasta may have introduced variability into the eating rate data, although this was adjusted for in the model. 

Strengths of this study were that participants were covertly timed and repeated measures of objective eating rate were made to obtain more reliable estimates. The meals were sized such that people were not uncomfortably full and the participants enjoyed the meals. This was important because the palatability of a meal is a variable in the determination of eating rate, with less favourable food being eaten more slowly [44]. 

## 5. Conclusions

The consistency of data with that in the published literature suggests that perceived speed of eating is sufficiently sensitive to detect group-level differences between fast and slow eating rates. At an individual level, low sensitivity and only fair agreement indicate that self-reported speed of eating is an unreliable method by which to assess an individual’s eating rate. Furthermore, there is no standardized protocol by which to assess speed of eating or eating rate. Establishing standardized protocols would enable the development of population reference ranges across various demographic groups that may be applicable for public health messages and in clinical management.

## Figures and Tables

**Figure 1 nutrients-12-00599-f001:**
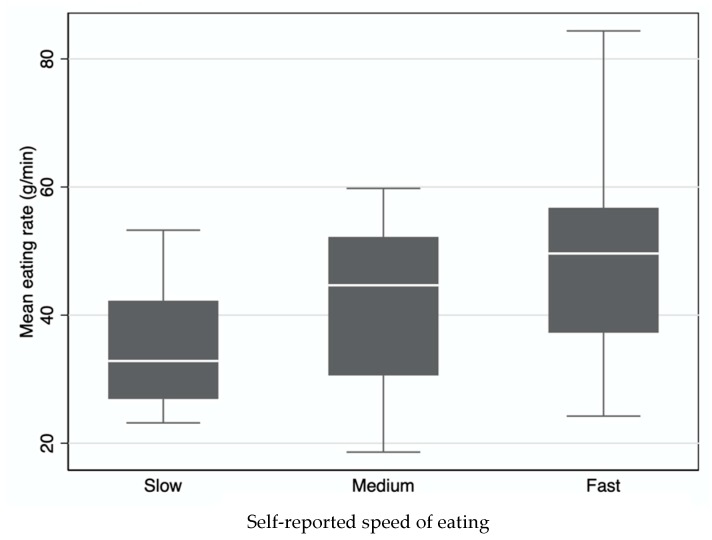
The medians (horizontal white lines) of the mean eating rate (g/min) among the self-classified speed of eating categories. The box represents the interquartile range (25 ^th^ to 75 ^th^ percentiles) and the whiskers give the full range of the mean eating rate.

**Table 1 nutrients-12-00599-t001:** Cooked weights (g) and energy content of the test meals.

Pasta or Rice	Non-Starchy Vegetables	Beef Mince and Bolognese Sauce	Energy (kJ)
100	250	200	1800
150	200	200	2000
200	150	200	2300

**Table 2 nutrients-12-00599-t002:** Participant demographics and objective eating rates by self-reported speed of eating category (*n* = 78).

Characteristic	Self-Reported Speed of Eating		
	Fast	Medium	Slow
N (%)	37 (47.4)	31 (39.7)	10 (12.8)
Age (y), mean (SD)	21.4 (2.0)	21.0 (2.6)	21.1 (0.7)
BMI (kg/m^2^), mean (SD)	23.6 (3.8)	23.2 (4.2)	24.3 (3.7)
Sex, *n* (%) female	26 (70.3)	25 (80.6)	7 (70.0)
NZ European ethnicity, n (%)	26 (70.3)	23 (74.2)	5 (50.0)
Objective eating rate, median (25th, 75th percentiles)			
Mean eating rate (g/min)	50 (37, 57)	45 (31, 52)	33 (27, 42)
Within-person SD (g/min)	6 (4, 9)	5 (3, 10)	4 (3, 8)
Coefficient of variation (%)	13 (8, 19)	14 (7, 24)	12 (10, 20)

Abbreviations: BMI, body mass index; NZ, New Zealand; SD, standard deviation.

**Table 3 nutrients-12-00599-t003:** Percentage agreement between self-reported and objective eating rate categories, *n* (%)**.**

Self-Reported Speed of Eating	Objective Eating Rate Groups		
	Slow (*n* = 10)	Medium (*n* = 31)	Fast (*n* = 37)
Slow (*n* = 10)	4 (5.1%)	4 (5.1%)	2 (2.6%)
Medium (*n* = 31)	5 (6.4%)	12 (15.4%)	14 (18.0%)
Fast (*n* = 37)	1 (1.3%)	13 (16.7%)	21 (26.9%)

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
