# Peer review of "Comparison of Self-Reported Speed of Eating with an Objective Measure of Eating Rate"

_nutrients, 2020, doi:10.3390/nu12030599_

Round 1
Reviewer 1 Report
This study compared self-reported speed of eating to 3 instances of objectively measured eating rate. This study’s contribution to the literature is the comparison of these measures. Strengths of this study include repeated instances of objectives measuring eating rate and statistical tests used to evaluate the comparison of self-report and objective measures. Overall, this article should address the weaknesses and comments discussed below.

Author Response
Reviewer comment: Some references are missing, these will be indicated by line below.
Author response: We have added references where indicated
Reviewer comment: Details in methods of self-report, eating habit questionnaire, and objective measurement sessions are missing.
Author response: We have rewritten the Methods to include these points
Reviewer comment: Discussion and conclusion(s) need deeper thinking about study results in relation to larger concerns (see first paragraph of introduction).
Author response: We have rewritten the Discussion
Reviewer comment: Eating, fast/slow eaters, eating rate, and speed of eating are used throughout, generally interchangeably. Suggest consistently using ‘speed of eating’ when referring to self-report and ‘eating rate’ when referring to objective measures.
Author response: Thank you, we have adopted your suggestion, much improved
Reviewer comment: The abstract covers the main aspects of the work and will spark interest in the right audience. However, it should be rewritten (specific suggestions below) to more clearly emphasize the results. It appears the main take home message is the relationship between the two measures, as such only those results with means (SD), CI, and p-values should be included.
Author response: We have rewritten the Abstract
Line 10: change ‘as to how to identify’ to ‘quantifying’ or ‘describing’
done
Lines 11-12: after 1-5 add (very slow – very fast) and delete sentence “The options….”
done
Lines 21-24: Suggest delete “in conclusion” and start sentence “Self- reported speed of eating…detect a significant difference…(p = 0.009); and fast… (p = 0.033). However, at an individual…. In addition, the range of self-reported speed of eating may be an unreliable… individual’s eating rate.
done
Introduction
Reviewer comment: The introduction is generally easy to follow but is missing information on objective studies of eating rate and citations.
Author response: We have added more detail with citations
Reviewer comment: 1st paragraph: a little reorganization is essential, suggest grouping related concepts eg. metabolic disease, BMI, etc.
Author response: This has been rewritten
Line 28: ‘domino effect’ is colloquial, used more refined language, this statement also requires a citation and reference
Reworded
Line 37: delete “which are on the rise worldwide”
Reworded
Line 43: change eating rate to speed of eating (self-report)
done
Line 44: delete “by which to judge the veracity of the claim”
reworded
Line 45: change eating rate to speed of eating (self-report)
done
Line 46: delete the “comparing the self-report with a friend report.”; change “In 222 subjects” to As a group, 222 subjects…
Line 48: change is to was; change subjective eating rate to self-report speed of eating
done
Lines 49-50: change ‘this paper…’ to ‘these results support the use of self-reported speed of eating.”
done
Line 51: add ‘objective’ before eating rate; need a discussion here or another paragraph of other’s finding in terms of objectives measures of eating rate; consider including evidence that led to the use of 3 times repeated eating occasions in the laboratory
Authors response: We have rewritten the Introduction to include these points
Materials and Methods
Reviewer comment: There are no major methodological concerns, but further details are needed for the general reader to be able to repeat the experiment. Creating subsections may help - General, Self-reported speed of eating, Objective measurement of eating rate, Statistical analysis.
Author response: We have added headings
Reviewer comment: General: how is this study a crossover design? There is no indication of different conditions. If this is crossover, significant details are missing, if not, please remove this term. The term washout may be used incorrectly, washout implies a need to remove something from the system, please explain or reword.
Author response: Thank you, this was not a crossover, it was a repeated experiment, we have amended.
Reviewer comment: Self-reported speed of eating: Was there an eating habits questionnaire as well (see Lines 105-106)? If there was just one question to describe self-report speed of eating, please make this clear; was this questionnaire given prior to every meal, so 3 times? How far in advance of the meal was this questionnaire given (consider attentional bias in discussion)? Was it on paper, verbal, electronic? Please provide more information about administration of self-reported speed of eating question and any other eating habits questionnaires
Author response: We have provided more detail in the Methods section Self-reported speed of eating
Reviewer comment: Objective measurement of eating rate: considerable details are missing about the food consumed and setting of consumption. While energy density is reported to be similar, total calories and macronutrient breakdown including fiber would be beneficial and should be a consideration. Was the researcher present in the room recording timing (in discussion consider social desirability bias)? If not, how were the subjects observed?
Author response: Yes, covertly timing with stopwatches
Reviewer comment: Were number of mouthfuls or size of mouthfuls considered? (in discussion consider those who may not have wanted to complete the meal or ‘shovelled’ food in early during eating then ate slower/smaller mouthfuls because they were required to eat everything)
Author response: Mouthfuls were not considered. We did not monitor eating behaviours (shovelling) or social interactions between students, we simply observed in order to record start and finishing times. Female dietetic students designed the meals to be of a manageable size and we have added this to the Methods.
Reviewer comment: Statistical analysis: explanations of how groups were created by self-report, objective, and combined and mixed between this section (or as another section after objective…) and results, consider introducing each of the groups here, then discussing in results, or just explaining each grouping in results.
Author response: Detail has been added to the Statistical analysis section of the Methods
ç The results provide all the necessary information to understand the basic details. However, reorganization would strengthen the results. Consider reorganizing results to align with methods – General, 3.1 Self-reported speed of eating, 3.2 Objective Eating Rate, and 3.3 Comparison or Relationship between … – currently some comparisons are mixed within each individual section. Reorganization will result in renumbering of Tables and Figures.
Author response: We have revised the presentation of the Results in response to your comments
Discussion
Reviewer comment: The discussion addresses the main finding and gives recognition to similar/related work. Consider reorganizing discussion to align with methods and results. 1st paragraph-overall, 2nd paragraph-comparison of findings to other selfreport findings; 3rd paragraph-comparison of objective results to other self-report objective results; 4th paragraph-comparison of relationship self-report/objective to others. Consider looking into the nursing literature for the benefits/risks of distracted mealtimes on eating (eg. https://doi.org/10.1136/bmj.332.7551.1165 this is just an example). Include in discussion items suggested in methods (above)
Author response: We have rewritten the Discussion
Reviewer comment: Line 148: the potential age range of the sample is 18-60, were they all ‘young’?
Author response: The mean age was 21y, there was one mature student.
Conclusions
Reviewer comment: Consider including ideas related to 1st paragraph of introduction, as may be appropriate
Author response: We have rewritten the Discussion
Line 214: delete our
Done
Line 216: change to ‘less certainty that self-report is sufficienctly…’
Removed
Tables and Figures
Reviewer comment: Table 1: consider: adding a column describing the sample overall, include n(%) for Female/Male and Race/Ethnicity; include p-values and effect sizes if there is any significant difference by speed of eating, if there are no significant differences, state this
Author response: We have added detail of the overall sample into text. The study was designed as within-person comparison between speed of eating and eating rate, not as a comparison between sexes or ethnicities. It would be interesting to design studies to compare between these characteristics but performing statistical tests in the present study could produce misleading information so we would prefer not to.
Reviewer comment: Figures 1 and 2: consider more descriptive titles
Author response: As part of our response to reviewers we have removed these figures
Reviewer 2 Report
Nutrients- 723286: Comparison of Self-reported Speed of Eating with an Objective Measure of Eating Rate
In this study, the authors compared self-reported eating speed to objective eating speed in a laboratory setting. They found that there was fair agreement between self-report and objective eating speeds; the authors conclude that self-reported eating speed is sufficient to detect group differences between those who eat fast vs. slow, but that there is poor sensitivity at the individual level. The authors should be commended on conducting a novel study with potential clinical importance for obesity prevention and intervention. There are a few confounding variables and limitations in the study design that should be addressed. In order to assist in addressing these limitations and strengthening the contributions of this study, the following considerations are offered:
Abstract
I am assuming there was not a significant difference between slow and medium eaters—could the authors please state this in the abstract? It would be helpful if the authors included a final sentence that describes the utility/importance of these results.
Introduction:
This sentence “Additionally, as eating quickly is associated with a higher BMI and central adiposity, there is an increased risk of developing type 2 diabetes” on p. 1 lines 34-36 should be rewritten or expanded upon. It is unclear whether there is a direct relationship between fast eating and type 2 diabetes risk or whether fast eating mediates the relationship between BMI/central adiposity and type 2 diabetes risk. The authors state on p. 2 lines 47-48 “However, at an individual level using a subjective measure (friend-report) is insufficient to validate a subjective eating rate question (self-report).” I agree with this point, but think that more justification is needed (e.g., the friend might also be a fast eater and have difficulty putting the participant’s rate of eating in context). Do the authors have a hypothesis about the degree of agreement between objective and self-reported measures, or is this study exploratory?
Methods/Procedures
Were participants alone when eating their meals? Were they allowed to look at their phones or do other tasks? Where were research staff located? Were all participants able to finish their meals? Why did the authors choose to have participants eat the entire meal instead of eating until they were full and then weighing the remaining food to calculate g/min? Based on participant size or appetite, eating the entirety of the meal may have been difficult and not similar to naturalistic settings. I am assuming that the authors collapsed the self-report categories such that “very slow” and “relatively slow” made up the “slow” category, etc. but this should be explicitly stated. Did the authors consider any covariates such as sex, BMI or age? While participants were asked to fast for 2.5 hours before the meal, did research staff record when their last meal was? This could greatly affect meal speed if one participant had voluntarily fasted for a day vs. another participant fasted exactly 2.5 hours.
Results:
Since age was pretty similar across subjective eating rate groups (mean of 21), how did the authors handle the participant who was 60 years old? Were any outliers dropped or transformed? This is especially relevant with the person who ate ~85g/min in Figure 2. What does the “Other” category refer to under “NZ European”? Although I realize this isn’t a central focus of this paper, were the objective rates of eating related to any of the correlates the authors mentioned in the introduction (e.g., BMI)? I wonder if there may be gender biases to report as a fast vs. slow eater (e.g., it may be seen as more masculine to report being a fast eater and more feminine to report being a slow eater). Were there differences in objective/subjective agreement by gender?
Discussion:
On p. 6, lines 193-194, “Interventions to reduce faster eating rate to 20-30g/min over a period of a year have been associated with significant weight loss in adolescents [35, 36].” I would clarify that this was among adolescents with obesity. I am glad that the authors mentioned the palatability of the meal. This should be mentioned earlier on in the Methods and Results sections. Did all participants rate the meal as “just right”? Was this adjusted for if someone rated it as not being “just right”? What were the options given to participants? The authors say that the participants were unaware that they were being timed. How was the study presented to the participants? For instance, did they think they were participating in a study about taste perception or about palatability?
General Comments:
Please be aware of grammatical errors throughout the manuscript. I noticed several.
Author Response
Abstract
Reviewer comment: I am assuming there was not a significant difference between slow and medium eaters—could the authors please state this in the abstract?
Author response: Now included
Reviewer comment: It would be helpful if the authors included a final sentence that describes the utility/importance of these results.
Author response: We have added a sentence
Introduction:
Reviewer comment: This sentence “Additionally, as eating quickly is associated with a higher BMI and central adiposity, there is an increased risk of developing type 2 diabetes” on p. 1 lines 34-36 should be rewritten or expanded upon. It is unclear whether there is a direct relationship between fast eating and type 2 diabetes risk or whether fast eating mediates the relationship between BMI/central adiposity and type 2 diabetes risk.
Author response: We have rewritten the Introduction
Reviewer comment: The authors state on p. 2 lines 47-48 “However, at an individual level using a subjective measure (friend-report) is insufficient to validate a subjective eating rate question (self-report).” I agree with this point, but think that more justification is needed (e.g., the friend might also be a fast eater and have difficulty putting the participant’s rate of eating in context). Do the authors have a hypothesis about the degree of agreement between objective and self-reported measures, or is this study exploratory?
Author response: The problem is that there is no means to assess speed of eating other than subjective perception, whether of oneself or by a third party. Also, there is no reference for eating rate, a rate that will depend on the type of food, palatability and social setting as well as the eating habits of the consumer. This lack of reference data hinder hypothesis testing at an individual level. We have rewritten the Discussion to address these issues.
Methods/Procedures
Reviewer comments: Were participants alone when eating their meals? Were they allowed to look at their phones or do other tasks? Where were research staff located? Were all participants able to finish their meals? Why did the authors choose to have participants eat the entire meal instead of eating until they were full and then weighing the remaining food to calculate g/min? Based on participant size or appetite, eating the entirety of the meal may have been difficult and not similar to naturalistic settings. I am assuming that the authors collapsed the self-report categories such that “very slow” and “relatively slow” made up the “slow” category, etc. but this should be explicitly stated. Did the authors consider any covariates such as sex, BMI or age? While participants were asked to fast for 2.5 hours before the meal, did research staff record when their last meal was? This could greatly affect meal speed if one participant had voluntarily fasted for a day vs. another participant fasted exactly 2.5 hours.
Author response: We have expanded the Methods to include these points
Results:
Reviewer comments: Since age was pretty similar across subjective eating rate groups (mean of 21), how did the authors handle the participant who was 60 years old? Were any outliers dropped or transformed? This is especially relevant with the person who ate ~85g/min in Figure 2. What does the “Other” category refer to under “NZ European”? Although I realize this isn’t a central focus of this paper, were the objective rates of eating related to any of the correlates the authors mentioned in the introduction (e.g., BMI)? I wonder if there may be gender biases to report as a fast vs. slow eater (e.g., it may be seen as more masculine to report being a fast eater and more feminine to report being a slow eater). Were there differences in objective/subjective agreement by gender?
Author response: Exclusion of the older participant did not affect results or inference so we have reported the whole sample. No outliers were dropped. We have included some more detail of the ethnic makeup in the Results section. The study was designed as within-person comparison between speed of eating and eating rate, not as a comparison between sexes or ethnicities. It would be interesting to design studies to compare between these characteristics but performing statistical tests in the present study could produce misleading information so we would prefer not to.
Reviewer comment: On p. 6, lines 193-194, “Interventions to reduce faster eating rate to 20-30g/min over a period of a year have been associated with significant weight loss in adolescents [35, 36].” I would clarify that this was among adolescents with obesity. I am glad that the authors mentioned the palatability of the meal. This should be mentioned earlier on in the Methods and Results sections. Did all participants rate the meal as “just right”? Was this adjusted for if someone rated it as not being “just right”? What were the options given to participants? The authors say that the participants were unaware that they were being timed. How was the study presented to the participants? For instance, did they think they were participating in a study about taste perception or about palatability?
Author response: We have substantially rewritten the Discussion and expanded the methods. The meals were designed by three dietetic students and the participants were asked to fill out a meal satisfaction questionnaire. They were unaware of being timed. We have expanded the Methods to include these details and described the scales used to record responses, including just right.
General Comments:
Please be aware of grammatical errors throughout the manuscript. I noticed several.
Thank you, we hope we have corrected these
Round 2
Reviewer 1 Report
The manuscript is greatly improved. Thank you. Moderate change is:
Lines 310-319: provide more details about the covert timing as well as the grouping of subjects (all the same participants each week?) this is partly discussed in limitations (lab v natural setting), please discuss further
Minor changes are as follows:
Line 229, m for weight should be kg
Line 334 should be assessed
Lines 595-597 condense to 1 sentence
Author Response
Reviewer comment: Lines 310-319: provide more details about the covert timing as well as the grouping of subjects (all the same participants each week?) this is partly discussed in limitations (lab v natural setting), please discuss further
Author response: We have added the following to the Methods
Participants were blinded to the eating rate component of the experiment, understanding that the purpose was to rate meal satisfaction.
The meals were served in two sittings, with participants allocated a start time of either 11:50 am or 1:00 pm. For each participant, the start time was the same for all three meals. Additionally, participants were block randomised by sex to the order in which they received the meals (either rice- or pasta-based) in a balanced manner such that there were approximately equal numbers of six meal order permutations.
Research staff were sat at desks offering clear visibility of their allocated participants with stopwatches concealed behind an upstand
Minor changes are as follows:
Line 229, m for weight should be kg
done
Line 334 should be assessed
done
Lines 595-597 condense to 1 sentence
We do not have line numbers this high so we were unable to identify these sentences